# Study of Crack-Propagation Mechanism of Al$_{0.1}$CoCrFeNi High-Entropy Alloy by Molecular Dynamics Method

**Cuixia Liu** [1,*] **and Yu Yao** [2]

1   School of Materials Science and Chemical Engineering, Xi'an Technological University, Xi'an 710021, China
2   Jiangsu Longda Superalloy Co., Ltd., Wuxi 214105, China
*   Correspondence: cuixialiu2016@sina.com; Tel.: +86-29-152-9145-7058

**Abstract:** The crack propagation mechanism of Al$_{0.1}$CoCrFeNi high-entropy alloy (HEA) was investigated with the molecular dynamics method. The pre-crack propagation and stretching processes of single-crystal Al$_{0.1}$CoCrFeNi HEA and Al$_{0.1}$CoCrFeNi HEA with grain boundaries were simulated. The effects of strain rates and different crystal structures on the crack propagation of the alloy therein at room temperature were studied. They both exhibited plastic deformation and ductile fracturing, and the crack tip involved dislocations at 45° and 135° under the tensile stress. The dislocations formed in the intrinsic-stacking fault and stacking fault based on hexagonal closely packed structures spread and then accumulated near the grain boundary. At the position where hexagonal closely packed structures were accumulated, the dent was obviously serious at the 1/3 position of the alloy where the fracturing finally occurred. The yield strength for Al$_{0.1}$CoCrFeNi HEA with grain boundaries was lower than that of the single-crystal Al$_{0.1}$CoCrFeNi HEA. However, Young's moduli for Al$_{0.1}$CoCrFeNi HEA with grain boundaries were higher than those of the single-crystal Al$_{0.1}$CoCrFeNi HEA. The grain boundaries can be used as the emission source of dislocations, and it is easier to form dislocations in the-single crystal Al$_{0.1}$CoCrFeNi HEA, but the existence of grain boundaries hinders the slippage of dislocations.

**Keywords:** high-entropy alloy; crack propagation; molecular dynamics; dislocation





## 1. Introduction

A high-entropy alloy (HEA) is multi-component alloy with equiatomic or near-equiatomic composition, which describes a solid solution composed of five or more metal elements. The molar ratios of these elements are similar or equal, usually ranging from 5% to 35% [1]. The excellent mechanical properties of Al$_{0.1}$CuFeNi HEA have attracted a large number of scholars to study it deeply because of its remarkable mechanical strength, oxidation resistance, and fatigue resistance at high temperatures [2,3]. In its plastic and elastic process, crack expansion is a typical mechanical behavior, one that is characteristic of HEAs, and it has an important influence on its mechanical properties. Cheng [4] simulated and calculated fracture toughness by molecular dynamics, which indicated that fracture toughness measured in stress intensity factor (or energy release rate) decreases with the decreasing crack length. The difficulties in preparing Al$_{0.1}$CrCuFeNi HEA prevented people from moving forward, and the mechanism of its plastic deformation is still unclear. Therefore, it is a significant breakthrough to study the microscopic evidence of the mechanical properties of Al$_{0.1}$CrCuFeNi HEA during crack expansion. During the plastic deformation of an HEA, near the crack tip, each dislocation nucleates and continues via emission. At the same time, stacking layers, or twins, are formed. Therefore, atomic shear behavior is caused by dislocation motion.

However, it is challenging to research the crack propagation mechanism for Al$_{0.1}$CrCuFeNi HEA. There are many factors that affect crack propagation, such as crystal structure, dislocation slip direction, elemental composition, and manufacturing processes. In early

work on the topic, our research team [5–7] discussed the effect of grain boundaries and short-range order on the mechanical properties of HEAs and explained the deformation mechanism. Zhang Z J et al. researched the local cracks on the surface and inside of an HEA with a FCC crystal structure [8]. The two cracks were roughly parallel to and extended along the same crystal direction; they were connected by a trace. The angle between the crack direction and the trace was about 71°, which is exactly the angle between the two (111) slip surfaces of HEA with FCC. The above results show that there is a certain relationship between crack propagation and dislocation slip in HEA with FCC.

Molecular dynamics is an effective method for exploring the essential mechanism of the mechanical properties of HEAs [9]. Li investigated $Co_{25}Ni_{25}Fe_{25}Al_{7.5}Cu_{17.5}$ HEAs by a series of molecular-dynamics tensile tests at different strain rates and temperatures [10]. Wang studied the mechanical behaviors and deformation mechanisms of scratched AlCrCuFeNi HEAs by molecular-dynamics simulations [11]. The movement trend of dislocation and stacking faults in crack expansion may be analyzed at the microscopic scale. In this study, molecular-dynamics was used to simulate the influence of loading rate on the propagation process of pre-cracks in an $Al_{0.1}CoCrFeNi$ HEA. The rest of paper is organized as follows: The description of the simulation method is offered in Section 2. Section 3 provides the discussion of the results of the simulation. The concluding remarks via the simulation results are given in Section 4.

## 2. Computational Methods

The interactions between atoms of matter is measured by a potential function which includes the embedded atom method (EAM) [12], Lennard–Jones potential [13], Mores potential [14], Johnson potential [15], and so on. Among them, the EAM potential has been applied to study the kinetics and solidification processes of liquid metal, surface structures, adsorption and microcosmic clusters, and many other things with great success. It can be expressed by the following:

$$E_{\mathrm{i}} = F_\alpha \left( \sum_{\mathrm{j} \neq i} \rho_\beta(r_{ij}) \right) + \frac{1}{2} \sum_{j \neq i} \phi_{\alpha\beta}(r_{ij}) \tag{1}$$

where, $E_{\mathrm{i}}$ is the sum of embedded energies and counter-potentials of two systems; $\alpha$ and $\beta$ represent the element types of atom $i$ and $j$; $r_{ij}$ is the distance between atom $i$ and $j$; $F$ represents the embedding energy function, which is the function of the electron density of all atoms in the system except itself $\rho$; $\Phi_{\alpha\beta}$ represents the potential interaction between elements $\alpha$ and $\beta$, which is a function of the distance $r_{ij}$ between atoms $i$ and $j$.

Based on the EAM theory, Professor Xia investigated the potential function of the AlCoCrFeNi system for several years and obtained an accurate potential function [16]. In this study, this potential function was used and also verified by us.

The models for $Al_{0.1}CoCrFeNi$ HEAs with single-crystal (SC-HEA) and $Al_{0.1}CoCrFeNi$ HEAs with grain boundaries (GB-HEA) are shown in Figure 1. Figure 1a shows a sketch map of a pre-crack for the SC-HEA. The standard of the International Organization for Standardization is not valid at the nanoscale, and therefore, the angle of the pre-crack may be random [17]. In this study, 45° was chosen. The size and orientation of the model in three-dimensional space (X, Y and Z) are given by 100a0 × 35a0 × 35a0; and $[11\bar{2}]$, [111], and $[1\bar{1}0]$, respectively. The lattice parameter a0 was 3.581Å. Finally, the SC-HEA was obtained as shown in Figure 1b,c. Compared with the SC-HEA, GB-HEA including grain boundaries (GB) contained a clustered microstructure, which was established based on the SC-HEA. The SC-HEA was uniformly heated from 300 to 2400 K and then relaxed at 2400 K in order to make all atoms keep a liquid state. After that, the $Al_{0.1}CoCrFeNi$ HEA was cooled down to room temperature at a certain cooling rate to form the cluster microstructure shown in Figure 1d,e. It can be seen that after solidification, $Al_{0.1}CoCrFeNi$ HEA still kept a FCC crystal structure. A small number of HCP and BCC atoms were precipitated after the polycrystalline model's relaxation by the crystal boundary.

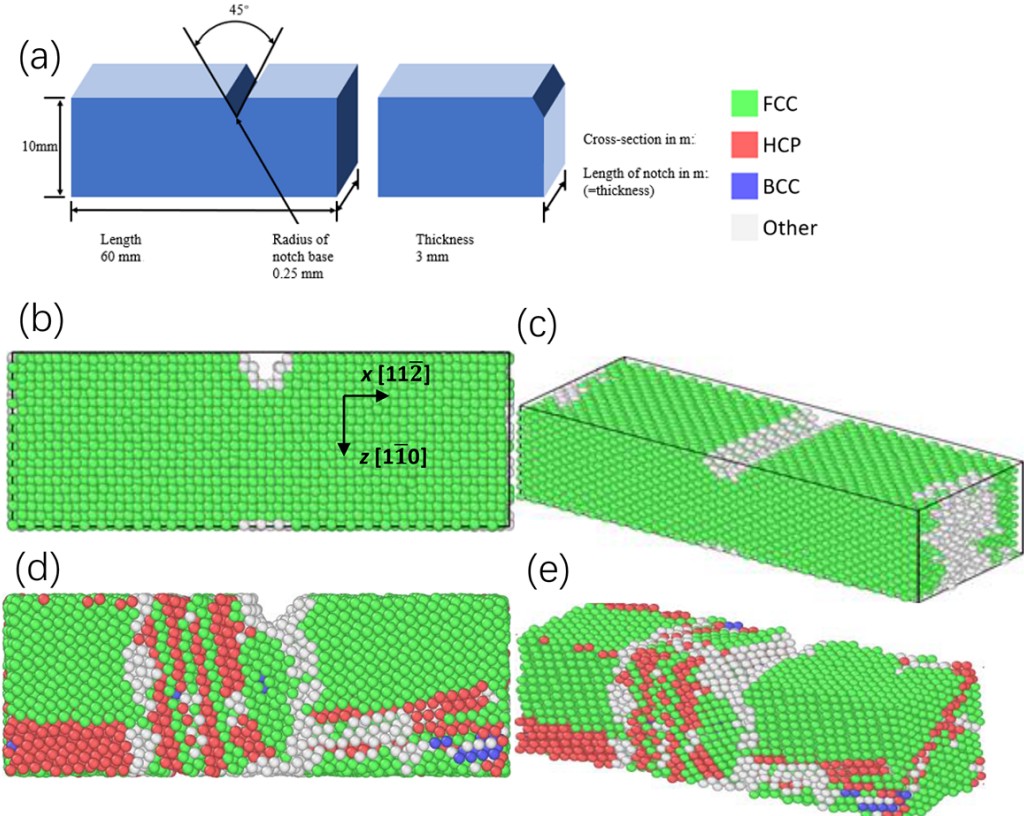

**Figure 1.** Model for a pre-crack in Al$_{0.1}$CoCrFeNi HEA—single crystal and grain boundary versions: (**a**) sketch map of pre-crack for SC-HEA; (**b**,**c**) the SC-HEA model; (**d**,**e**) the GB-HEA model.

## 3. Results and Discussion

Based on above models of SC-HEA and GB-HEA, the crack's direction is $(11\overline{2})\left[1\overline{1}0\right]$. It is important to choose an appropriate strain rate when Al$_{0.1}$CrCuFeNi HEA is stretched in the molecular dynamics method. If the strain rate is too low, it would require a very long simulation time. If the strain rate is too high, it would result in higher tensile strength. Therefore, the strain rate should not only make valence bonds between atoms fractured, but also take into account the time allowed for the computer simulation. If the strain rate is less than $10^8$ s$^{-1}$, it takes too long to simulate. If it is higher than $10^{10}$ s$^{-1}$, the tensile strength is too high. In this paper, SC-HEA and GB-HEA were stretched at a constant quasi-static loading speed along the x axis with different strain rates, including $5 \times 10^8$, $8 \times 10^8$ s$^{-1}$, $1 \times 10^9$, and $2 \times 10^9$ s$^{-1}$ and $5 \times 10^9$ s$^{-1}$ at 300 K in order to investigate the microstructural evolution of crack propagation.

### 3.1. Crack Propagation for SC-HEA

The evolution of crack propagation microstructurally in SC-HEA is shown in Figure 2. Overall, the deformation processes of crack propagation in SC-HEA with different strain rates are similar. Take the deformation process of $8 \times 10^8$ s$^{-1}$ (Figure 2b) as an example: Before $\varepsilon$ = 0.034, it is in the initial stage of crack propagation; the applied load has not yet acted on the crack-tip area, and the concentrated stress on the crack tip has not yet reached the critical value, resulting in the crack width increasing along the x axis and the length remaining unchanged. It can be seen in Figure 2b that when $\varepsilon$ = 0.034, the crack tip begins to propagate, and the critical stress at the crack tip is 5.45 GPa (shown in Figure 3b). The arrow in the figure shows the first emitted dislocation. At this time, FCC is the main feature of the atomic crystal structure of Al$_{0.1}$CoCrFeNi HEA. When $\varepsilon$ > 0.034, the crack tip begins to distort gradually, and the crack starts to propagate rapidly along the [1$\overline{1}$0] direction. When $\varepsilon$ = 0.053, the atoms near the crack tip (area A) move in disorder. As the crack tip

emits dislocations many times, passivation occurs to a certain extent, resulting in the stress always being concentrated on the crack tip. The dislocations emitted from the crack tip move along the direction of the slip plane in Figure 2b, and a dislocation-free area (area B) is formed between the movement track and the crack tip. The stress is redistributed at the front end of the crack tip after the dislocation is emitted. The new deformation mechanisms appear in the crack-tip-area dislocation and dislocation ring, which greatly release the stress concentration phenomenon into the system. When $\varepsilon = 0.085$, due to the continuous emission of dislocations, there are layered, hexagonal, closely packed (HCP) atoms (red atoms) and a small number of body-centered cubic (BCC) atoms (blue atoms). The HCP structure may form a stacking fault (SF). When $\varepsilon = 0.304$, the necking phenomenon appears gradually, which indicates that the deformation process of SC-HEA belongs to plastic deformation. After that, the SC-HEA enters an uneven plastic deformation stage, during which twins and stacking faults keep growing and disappearing. The system is about to break when $\varepsilon$ is 0.521. It is judged that SC-HEA exhibits ductile fracturing [18]. When the strain rate increases from $5 \times 10^8$ s$^{-1}$ to $5 \times 10^9$ s$^{-1}$, when the fracture occurs, the strain value ($\varepsilon$) fluctuates but mostly reduces, giving values such as 0.630, 0.521, 0.540, 0.512, and 0.533. In the process of crack propagation, the stress always concentrates on the crack tip, which leads to obvious passivation of the crack tip. Dislocations are intermittently emitted from the crack tip along two symmetrically distributed slip surfaces, and the inclination angles of the two symmetrically distributed slip surfaces are 45° and 135°, shown in Figure 2. The dislocation moves forward along the slip plane to pile up at the location of boundary, which causes a great stress concentration where dislocations accumulate. As the stretching continues to increase, the dislocation will cross the boundary and extend into neighboring grains. Therefore, plastic deformation would occur until the SC-HEA fractures. At the same time, vacancies would be formed and the extrinsic stacking fault (ESF) will appears, as shown in Figure 2b,c,e. The structure for ESF is always that two HCP layers are sandwiched between FCC layers during deformation, which is consistent with the theoretical analysis [19] and the experimental results [20]. During the plastic deformation of AlCoCrFeNi$_{2.1}$ HEA in an in situ tensile test monitored by a transmission electron microscope, dislocations were emitted from FCC phase and piled up in large quantities at the right phase boundary.

The stress–strain curves of SC-HEA under different strain rates are shown in Figure 3. It can be seen that there is an initial linear elastic stage at first, and a linear equation was fitted; see Figure 3. The stress reached its peak, and then the stress dropped sharply until fracture. After that, the deformation of SC-HEA entered the plastic deformation stage. With increasing strain, SC-HEA continued deform until it fractured. When the strain rate increased from $5 \times 10^8$ to $5 \times 10^9$ s$^{-1}$, the strain $\varepsilon$ corresponding to the peak value of yield stress was 0.0423, 0.0550, 0.0406, 0.0371, or 0.0432, respectively, and the speed of the peak value in the tensile process of the system fluctuated.

Table 1 lists the changes in yield strength and Young's modulus with strain rate. It can be observed that with an increase in strain rate, the yield strength and Young's modulus both increase, although they had little fluctuation. This indicates that a higher strain rate for SC-HEA will cause higher yield strength and a higher Young's modulus.

**Table 1.** Young's modulus and yield strength of SC-HEA at different strain rates.

| Strain Rate/s$^{-1}$ | $5 \times 10^8$ | $8 \times 10^8$ | $1 \times 10^9$ | $2 \times 10^9$ | $5 \times 10^9$ |
|---|---|---|---|---|---|
| Yield strength/GPa | 5.7786 | 5.9517 | 5.7458 | 6.1128 | 6.4878 |
| Young's modulus/GPa | 66.8081 | 52.4974 | 69.2321 | 89.7900 | 74.7483 |

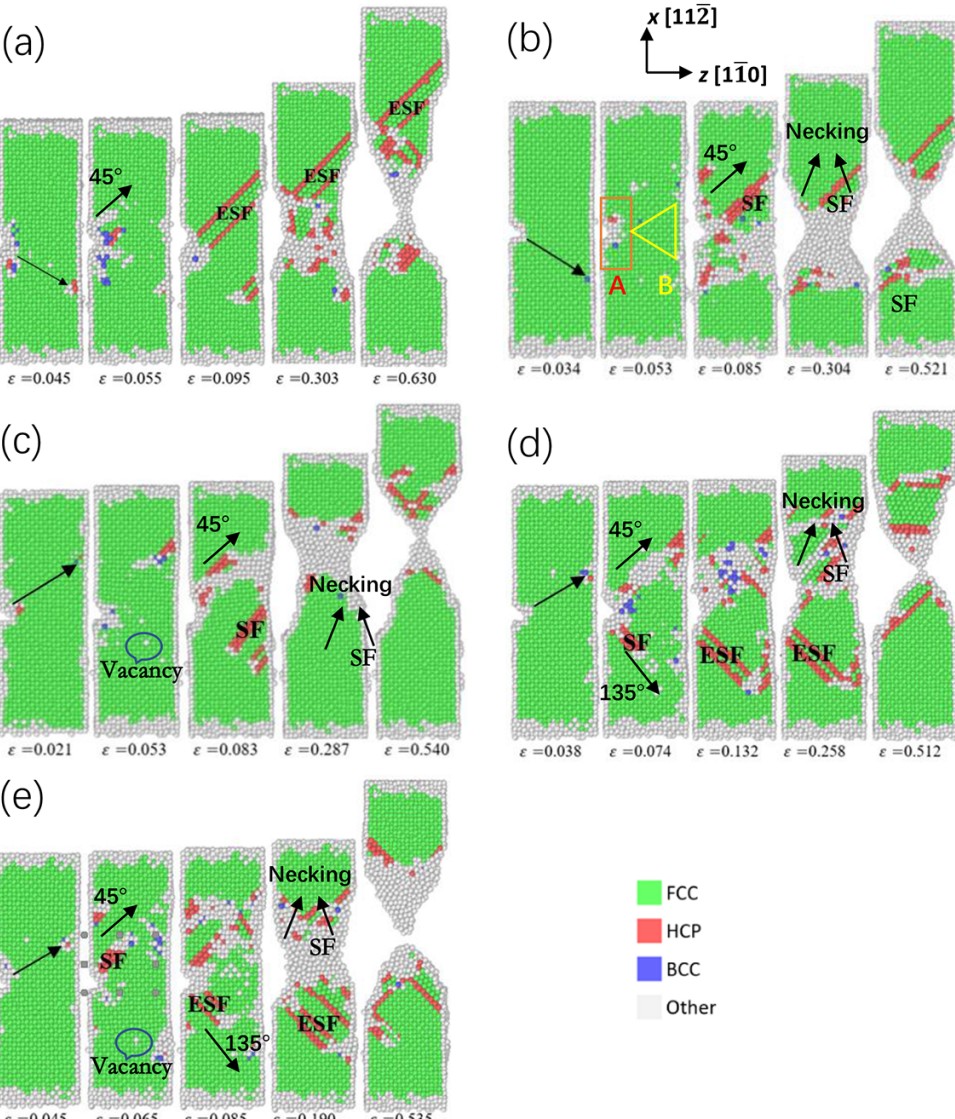

**Figure 2.** Crack propagation process of SC-HEA in tensile tests at different strain rates: (**a**) $5 \times 10^8$ s$^{-1}$ (**b**) $8 \times 10^8$ s$^{-1}$ (**c**) $1 \times 10^9$ s$^{-1}$ (**d**) $2 \times 10^9$ s$^{-1}$ (**e**) $5 \times 10^9$ s$^{-1}$.

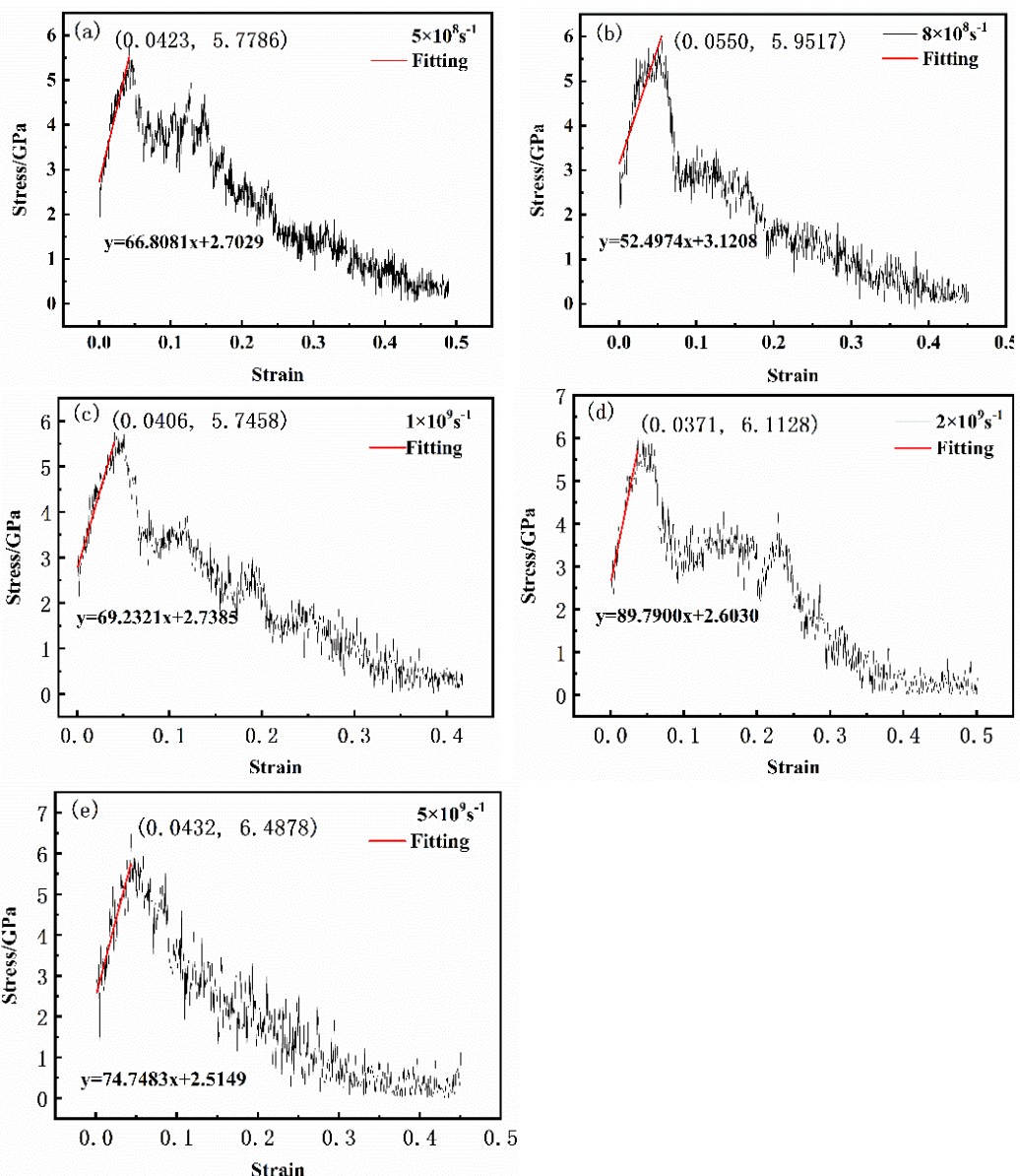

**Figure 3.** Stress–strain curves of SC-HEA at different strain rates at 300 K: (**a**) $5 \times 10^8$ s$^{-1}$ (**b**) $8 \times 10^8$ s$^{-1}$ (**c**) $1 \times 10^9$ s$^{-1}$ (**d**) $2 \times 10^9$ s$^{-1}$ (**e**) $5 \times 10^9$ s$^{-1}$.

### 3.2. Crack Propagation for GB-HEA

Compared with the crack propagation process of SC-HEA, that of GB-HEA is shown in Figure 4. The deformation process of GB-HEA is similar to that of SC-HEA, which is also plastic deformation. However, GB-HEA contains GB, and the motion of the dislocation is clearly affected by GB. Take the deformation process of $8 \times 10^8$ s$^{-1}$ (Figure 4b) as an example. In Figure 4b, when $\varepsilon = 0.034$, because there are GB under the crack tip, the strain field expands from the crack tip and interacts with the GB. No dislocation occurs at the crack tip, which proves GB may impede the propagation of cracks. When $\varepsilon = 0.053$, dislocations are emitted from the GB (area A). When $\varepsilon = 0.085$, deformations at area A continue to extend to the interior of HEA slowly. At the crack tip, the GB structure still hinders dislocation, which results in more dislocations at the GB. SF are formed in the middle area. When $\varepsilon = 0.304$, the strain field at the crack tip continues to act with the GB, and the stress concentration phenomenon occurs. The number of disordered atoms (white atoms) increases with the amount of deformation. At this time, the slip band will find it difficult to alleviate this stress concentration phenomenon, resulting in serious damage

to the internal structure. This shows that the depression of GB-HEA is obviously serious at the place where the HCP structure accumulates in the up 1/3 position. The intrinsic stacking fault (ISF) is formed during the tensile process, which is shown as two adjacent HCP crystal planes when $\varepsilon = 0.304$. When $\varepsilon = 0.521$, the system shows an obvious necking phenomenon. The crack tip gradually merges with the GB. The twin boundary (TB) appears as a single-layer HCP crystal structure. In general, TB is achieved by moving adjacent atomic planes based on the ISF structure. After the ISF structure is blocked during the extension, in order to ensure the continuous improvement of material plasticity, the ISF develops into a TB. On the one hand, the TB adjusts the crystal orientation, and then further stimulates the structure slip. On the other hand, the twin boundary reduces the average free path of the dislocation and enhances the strain hardening and improves the plasticity, which is consistent in with the TWIP effect in reference [21].

According to Figure 4, when the strain rate increases from $5 \times 10^8$ to $5 \times 10^9$ s$^{-1}$, when the fracture occurs, the strain ($\varepsilon$) fluctuates—0.575, 0.617, 0.705, 0.662, and 0.630, respectively, most of these values being higher than those of SC-HEA.

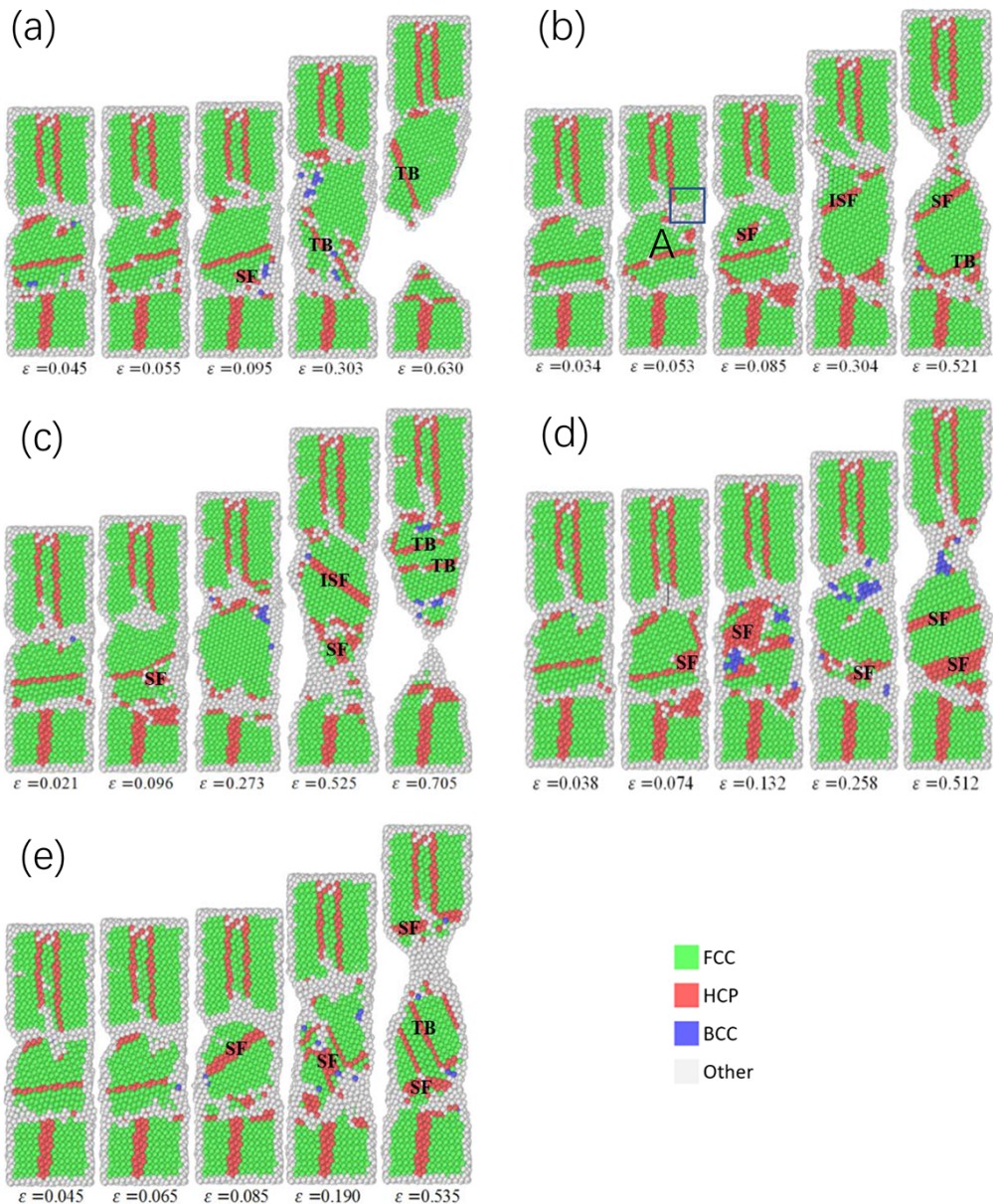

**Figure 4.** Crack propagation process of cracked GB-HEA in tensile tests with different strain rates: (**a**) $5 \times 10^8$ s$^{-1}$ (**b**) $8 \times 10^8$ s$^{-1}$ (**c**) $1 \times 10^9$ s$^{-1}$ (**d**) $2 \times 10^9$ s$^{-1}$ (**e**) $5 \times 10^9$ s$^{-1}$.

The stress–strain curves of GB-HEA under different strain rates are shown in Figure 5. In relation to the stress–strain curves of SC-HEA, they have the same variation trends. They both go through a linear elastic region to the peak stress, and the stress drops rapidly after reaching the peak. After that, they enter the plastic deformation stage. When the strain rate increases from $5 \times 10^8$ to $5 \times 10^9$ s$^{-1}$, the strain $\varepsilon$ when the yield stress reaches the peak value is 0.0302, 0.0362, 0.0350, 0.0378, or 0.0361, respectively.

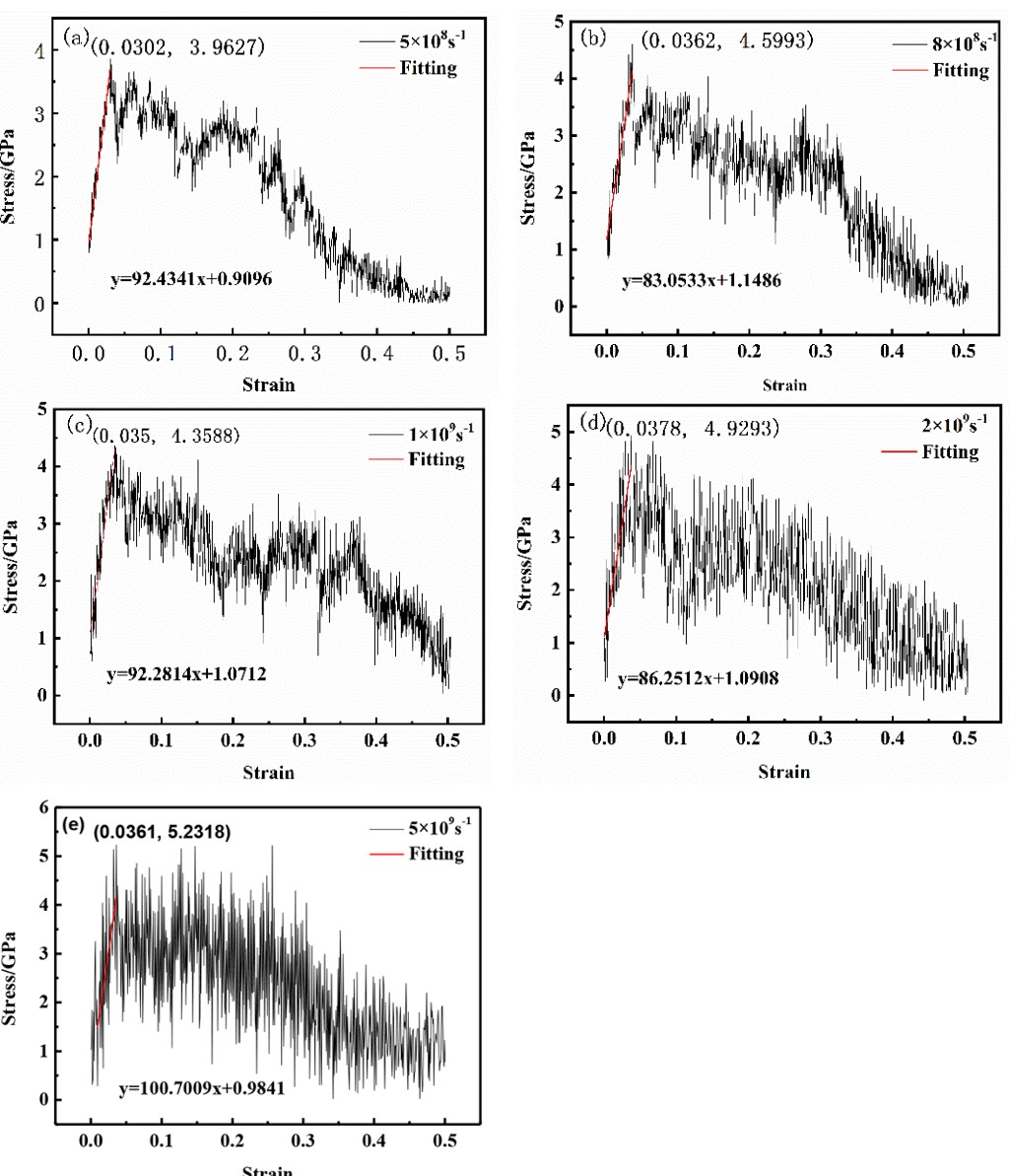

**Figure 5.** Stress–strain curves of GB-HEA at different loading rates at 300 K: (**a**) $5 \times 10^8$ s$^{-1}$ (**b**) $8 \times 10^8$ s$^{-1}$ (**c**) $1 \times 10^9$ s$^{-1}$ (**d**) $2 \times 10^9$ s$^{-1}$ (**e**) $5 \times 10^9$ s$^{-1}$.

At the same time, the yield strengths for GB-HEA shown in Table 2 are lower than those of SC-HEA. However, the Young's moduli for GB-HEA are higher than those of SC-HEA, although the Young's modulus for GB-HEA fluctuated a little. Those characteristics all prove that GB may hinder the movement of dislocation. The yield strengths and Young's moduli are compared in Figure 6. Compared with SC-HEA, GB can be used as the emission source of dislocations, which means that by GB, the number of initial slip systems increases, the plastic deformation capacity of the material increases, the stress required for material yield decreases, and the yield strength also decreases. On the other hand, the Young's moduli of the GB-HEA are higher than those of SC-HEA.

**Table 2.** Young's modulus and yield strength of GB-HEA at different strain rates.

| Strain Rate/s$^{-1}$ | $5 \times 10^8$ | $8 \times 10^8$ | $1 \times 10^9$ | $2 \times 10^9$ | $5 \times 10^9$ |
|---|---|---|---|---|---|
| Yield strength/GPa | 3.9627 | 4.5993 | 4.3588 | 4.9293 | 5.2318 |
| Young's modulus/GPa | 92.4341 | 83.0533 | 92.2814 | 86.2512 | 100.7009 |

**Figure 6.** Comparison of Young's modulus and yield strength between single crystal Al$_{0.1}$CoCrFeNi HEA and Al$_{0.1}$CoCrFeNi HEA with grain boundary.

## 4. Conclusions

SC-HEA and GB-HEA were investigated under different strain rates. The crack propagation mechanisms were analyzed, and the results show that that they both exhibit plastic deformation and ductile fracturing during the tensile process. Dislocations are emitted during the propagation process, the crack tip involves passivation, and the dislocations emitted from the crack tip accumulate near the GB. With the increase in strain rate, the crack propagation rate slows down and the plastic strength of the material increases. The cracks of Al$_{0.1}$CoCrFeNi HEA emit dislocations along the directions of 45° and 135° under the tensile stress. The ISF and SF based on HCP were observed during the deformation process. The fracture finally occurs obviously at the 1/3 position. With the increase in strain rate, the yield strengths and Young's moduli for the SC-HEA and GB-HEA fluctuated, but the overall trends were still increasing. The yield strength for GB-HEA was lower than that of SC-HEA. However, the Young's moduli for GB-HEA were higher than those for SC-HEA. The existence of GB can be used as the emission source of dislocations, and it is easier to form dislocations than SC-HEA, but the existence of GB hinders the slippage of dislocations.

**Author Contributions:** C.L.: methodology, software, validation, investigation, visualization, writing—original draft, writing—review and editing. Y.Y.: software, writing—review and editing. All authors have read and agreed to the published version of the manuscript.

**Funding:** This research was funded by the Natural Science Foundation of China (grant number 51971166) and Shaanxi Provincial Science and Technology Plan Project (no. 2021JM-430).

**Institutional Review Board Statement:** Not applicable.

**Informed Consent Statement:** Not applicable.

**Data Availability Statement:** Data is contained within the article. The data presented in this study can be seen in the content above.

**Conflicts of Interest:** The authors declare no conflict of interest.

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
