# Peer review of "Study of Crack-Propagation Mechanism of Al0.1CoCrFeNi High-Entropy Alloy by Molecular Dynamics Method"

_crystals, doi:10.3390/cryst13010011_

Round 1

Reviewer 1 Report

The manuscript is relevant to the field and well structured. More than half of the cited papers are older than 5 years. There is no self-citation. The conclusions are consistent with the results.

1. There are no references to [12], [14], [18] in the text.

2. The author points out the complexity of studying the mechanism of crack propagation in high-entropy alloys, further notes that molecular dynamics is an effective method for assessing the mechanisms of mechanical properties of these materials. At the same time, it is not clear from the review whether other authors have previously performed similar simulations for an alloy of this composition. What is the difference from [13].

3. Why was this composition chosen?

4. It is not clear why such strain rates were chosen.

5. Unfortunately, figures 2 and 4 are too low resolution. This does not allow tracking all the points that are indicated in the text in Section 3, as well as checking the results formulated on the basis of these figures. It is also difficult to understand the labels themselves in these figures (for example, the value of ε).

6. What are "other atoms"? Only on line 174 does it appear that these are passivating atoms. What does the author mean in this case?

7. Why is the model with a grain boundary formed in such a way that the precrack is exactly at the grain boundary?

8. Lines 122 and 123 state:

"When the strain rate increases from 5 × 121 108 s−1 to 5 × 109 s−1, when the fracture occurs, the strain is obviously reduced such as ε is 0.630, 0.521, 0.540, 0.512 and 0.533 respectively"

However, the value fluctuates rather than decreases.

9. The appearance of a grain boundary in a single crystal (line 128) is not described in any way.

10. The author in Figure 3 gives the result from [17]. However, the presence of these phases has not been confirmed in any way (for example, by diffraction). This work is not presented in databases and is difficult to find for confirmation.

11. An explanation is not given in which cases the layers were determined as the boundaries of a twin, and in which cases as a stacking fault.

For example, lines 180-181:

"The crack tip gradually merges with the GB. The twin boundary (TB) appears shown as single-layer HCP crystal structure"

At the same time, in the same figure, a single-layer HCP crystal structure is marked as a stacking fault.

12. Line 183 states:

"...reduced such as 0.575, 0.617, 0.705, 0.662 and 0.630 respectively..."

however, such data cannot be seen in the figure.

Author Response

Response to Reviewer 1 Comments

The manuscript is relevant to the field and well structured. More than half of the cited papers are older than 5 years. There is no self-citation. The conclusions are consistent with the results.

Point 1: There are no references to [12], [14], [18] in the text.

Response 1:

  All references in this text have been carefully checked and adjusted. In addition, references [5], [6],[7] and [9] belong to our research team’s findings.

Point 2: The author points out the complexity of studying the mechanism of crack propagation in high-entropy alloys, further notes that molecular dynamics is an effective method for assessing the mechanisms of mechanical properties of these materials. At the same time, it is not clear from the review whether other authors have previously performed similar simulations for an alloy of this composition. What is the difference from [13].

Response 1:

Other authors also have done similar studies on high entropy alloys (HEAs), such as Li [10] and Wang [11]. Li investigated Co25Ni25Fe25Al7.5Cu17.5 HEAs by a series of molecular-dynamics tensile tests at different strain rates and temperatures. Wang studied the mechanical behaviors and deformation mechanisms of scratched AlCrCuFeNi HEAs by molecular dynamics simulations. However, for this composition (AlxCoCrFeNi), it is still relatively few, because of the lack of an exact potential function. In our work of “(Reference 16, same to 13 in the first manuscript), we mainly researched the effect of local atomic configuration in high-entropy alloys on the dislocation behaviors and mechanical properties, we finally find a gap between local atomic configuration (LAC) and mechanical properties for HEAs by associating the influence of LAC to the dynamic scope. And what's more, an accurate potential function for AlxCoCrFeNi HEAs was developed based on EAM and validated carefully through some bulk properties (Lattice and elastic constants) and defect energies (vacancy energy, cohesive energy, twin energy and stacking fault energy) obtained by the first-principle calculation and experimental work)

At the same time, we revised the line 56-60 in our manuscript.

“Li investigated Co25Ni25Fe25Al7.5Cu17.5 HEAs by a series of molecular-dynamics tensile tests at different strain rates and temperatures [10]. Wang studied the mechanical behaviors and deformation mechanisms of scratched AlCrCuFeNi HEAs by molecular dynamics simulations [11].”

Point 3: Why was this composition chosen?

Response 3:

There are two reasons. Firstly, AlxCrCuFeNi alloy belongs to HEA, which containing at least four components without major elements. They have excellent mechanical properties, good high-temperature stability and high resistance to wear and corrosion. However, the difficulty of the preparation prevented people from moving forward and the mechanism of plastic deformation are still unclear. We hope this study could help researchers to increase the properties of HEA.

Secondly, we intend to use the molecular dynamics method, in which the potential function is very significant and necessary. Moreover, our research team has investigated the potential function of this composition for several years and obtained an accurate potential function. For other composition HEA, it is difficult to find exact potential functions.

At the same time, we revised it in our manuscript.

line 30-32: “The excellent mechanical properties of AlxCrCuFeNi HEA have attracted a large number of scholars to study deeply because of its remarkable mechanical strength, oxidation resistance and fatigue resistance at high temperature [2,3]..”

line 37-38: “The difficulty of preparation for AlxCrCuFeNi HEA prevented people from moving forward and the mechanism of plastic deformation are still unclear.”

line 80-81 : “Based on the EAM theory, Professor Xia has investigated the potential function of AlCoCrFeNi system for several years and obtained an accurate potential function [16].”

Point 4: It is not clear why such strain rates were chosen.

Response 4:

It is important to choose an appropriate strain rate when Al0.1CrCuFeNi HEA is stretched in the molecular dynamics method. If the strain rate is too low, it would take a very long simulation time. If the strain rate is too high, it would result in higher tensile strength. Therefore, strain rate should not only make valence bond between atoms fractured, but also limit within the allowed time of the computer simulation. In this paper, strain rate includes 5×108 s-1,8×108 s-1,1×109 s-1,2×109 s-1, 5×109 s-1. If strain rate is less than 108 s-1, it takes too long simulation time. If it is higher than109 s-1, the tensile strength is too high.

At the same time, we revised it in our manuscript.

line 103-109:” It is important to choose an appropriate strain rate when Al0.1CrCuFeNi HEA is stretched in the molecular dynamics method. If the strain rate is too low, it would take a very long simulation time. If the strain rate is too high, it would result in higher tensile strength. Therefore, strain rate should not only make valence bond between atoms fractured, but also limit within the allowed time of the computer simulation. If strain rate is less than 108 s-1, it takes too long simulation time. If it is higher than1010 s-1, the tensile strength is too high.”

Point 5: Unfortunately, figures 2 and 4 are too low resolution. This does not allow tracking all the points that are indicated in the text in Section 3, as well as checking the results formulated on the basis of these figures. It is also difficult to understand the labels themselves in these figures (for example, the value of ε).

Response 5:

All figures including figure 2 and 4 have been revised carefully according to the requirement of this journal.

Point 6: What are "other atoms"? Only on line 174 does it appear that these are passivating atoms. What does the author mean in this case?

Response 6:

We revised the line 185-195 in our manuscript.

“when ε = 0.034, because there is GB under the crack tip, the strain field expands from the crack tip and interacts with the GB. No dislocation occurs at the crack tip, which proves GB may impede the propagation of crack. With the loading continues. When ε = 0.053, dislocations are emitted from the GB (area A).When ε = 0.085, deformation at area A continue to extend to the interior of HEA slowly. At crack tip, the GB structure is still hinder of dislocation, which result in more dislocations at the GB. SF are formed in the middle area. When ε = 0.304, the strain field at the crack tip starts to continue to act with the GB and stress concentration phenomenon is formed. The number of disordered atoms (white atoms) increases with the amount of deformation. At this time, the slip band is difficult to alleviate this stress concentration phenomenon, resulting in serious damage to the internal structure..”

Point 7: Why is the model with a grain boundary formed in such a way that the precrack is exactly at the grain boundary?

Response 7:

Compared with the single crystal (SC-HEA), Al0.1CoCrFeNi HEAs with grain boundary (GB-HEA) including grain boundary (GB) contains cluster microstructure, which is established based on the SC-HEA. SC-HEA was uniformly heated from 300 K to 2400 K, and then relaxed at 2400 K in order to make all atoms to keep liquid state. After that the Al0.1CoCrFeNi HEA was cooling down to room temperature at a certain cooling rate to form the cluster microstructure shown in Figure 1(d,e). This structure containing cluster microstructure is called GB-HEA.

The precrack is exactly at the grain boundary, the reason of which is that we got many models through solidifying SC-HEA. We choose one of those models and guarantee this structure.

Point 8: Lines 122 and 123 state:

"When the strain rate increases from 5 × 121 108 s−1 to 5 × 109 s−1, when the fracture occurs, the strain is obviously reduced such as ε is 0.630, 0.521, 0.540, 0.512 and 0.533 respectively"

However, the value fluctuates rather than decreases.

Response 8:

We revised the line 139-141 in our manuscript.

“When the strain rate increases from 5 × 108 s−1 to 5 × 109 s−1, when the fracture occurs, the strain value (ε) fluctuates and mostly reduced such as 0.630, 0.521, 0.540, 0.512 and 0.533 respectively.”

Point 9: The appearance of a grain boundary in a single crystal (line 128) is not described in any way.

Response 9:

We revised the line 145-153 in our manuscript.

“The dislocation moves forward along the slip plane to pile up at the location of boundary, which causes a great stress concentration where dislocation accumulate. As the stretching continues to increase, the dislocation would cross the boundary and extend into neighboring grains. Therefore, plastic deformation would occur until the SC-HEA fractures. At the same time, vacancy would be formed and the extrinsic stacking fault (ESF) appears shown in Figure 2(b,c,e). The structure for ESF is always that two HCP layers are sandwiched with FCC layers during deformation, which is consistent with the theoretical analysis [19] and the experimental results [20] in Figure 3.”

Point 10: The author in Figure 3 gives the result from [17]. However, the presence of these phases has not been confirmed in any way (for example, by diffraction). This work is not presented in databases and is difficult to find for confirmation.

Response 10:

We want to explain the plastic deformation of HEAs was dominated by dense dislocation activities in FCC phase where dislocation were emitted from phase boundaries, which is coincident with that of Reference [20] (same to [17] in the first manuscript). You are right, we need to confirm it by some test method.

Point 11: An explanation is not given in which cases the layers were determined as the boundaries of a twin, and in which cases as a stacking fault.

For example, lines 180-181:

"The crack tip gradually merges with the GB. The twin boundary (TB) appears shown as single-layer HCP crystal structure"

At the same time, in the same figure, a single-layer HCP crystal structure is marked as a stacking fault.

Response 11:

In general, TB is achieved by moving adjacent atomic planes based on ISF structure. After ISF structure is blocked during the extension, in order to ensure the continuous improvement of material plasticity, the ISF is developed into TB. On the one hand, the TB adjusts the crystal orientation, and then further stimulates the structure slip. On the other hand, the twin reduces the average free path of the dislocation and enhances the strain hardening and improves the plasticity, which is consistent in with TWIP effect in reference [20].

We also revised them and explain them carefully in our manuscript.

line 132-135 “When ε = 0.085, due to the continuous emission of dislocations, it is found that layered hexagonal close packed (HCP) atoms (red atoms), a small number of body-centered cubic (BCC) atoms (blue atoms). HCP structure may form stacking fault (SF).”

line 145-153 “The dislocation moves forward along the slip plane to pile up at the location of boundary, which causes a great stress concentration where dislocation accumulate. As the stretching continues to increase, the dislocation would cross the boundary and extend into neighboring grains. Therefore, plastic deformation would occur until the SC-HEA fractures. At the same time, vacancy would be formed and the extrinsic stacking fault (ESF) appears shown in Figure 2(b,c,e). The structure for ESF is always that two HCP layers are sandwiched with FCC layers during deformation, which is consistent with the theoretical analysis [19] and the experimental results [20] in Figure 3.”

line 197-206 “The intrinsic stacking fault (ISF) is formed during the tensile process, which is shown as two adjacent HCP crystal planes when ε = 0.304. When ε = 0.521, the system shows an obvious necking phenomenon. The crack tip gradually merges with the GB. The twin boundary (TB) appears shown as single-layer HCP crystal structure. In general, TB is achieved by moving adjacent atomic planes based on ISF structure. After ISF structure is blocked during the extension, in order to ensure the continuous improvement of material plasticity, the ISF is developed into TB. On the one hand, the TB adjusts the crystal orientation, and then further stimulates the structure slip. On the other hand, the twin reduces the average free path of the dislocation and enhances the strain hardening and improves the plasticity, which is consistent in with TWIP effect in reference [21].”

Point 12: Line 183 states:

"...reduced such as 0.575, 0.617, 0.705, 0.662 and 0.630 respectively..."

however, such data cannot be seen in the figure.

Response 12:

For SC-HEA, all of those strain (ε) value for 0.630, 0.521, 0.540, 0.512 and 0.535 could be found under last structure in (a,b,c,d,e) of Figure 2.

For GB-HEA, all of those strain (ε) value for 0.575, 0.617, 0.705, 0.662 and 0.630 could be found under last structure in (a,b,c,d,e) of Figure 5.

We also revised them in our manuscript.

Line 139-141 For SC-HEA, “When the strain rate increases from 5 × 108 s−1 to 5 × 109 s−1, when the fracture occurs, the strain value (ε) fluctuates and mostly reduced such as 0.630, 0.521, 0.540, 0.512 and 0.533 respectively.”

Line 209-211 For GB-HEA,” According to Figure 5, when the strain rate increases from 5 × 108 s−1 to 5 × 109 s−1, t when the fracture occurs, the strain (ε) fluctuates such as 0.575, 0.617, 0.705, 0.662 and 0.630 respectively, which is higher mainly than that of SC-HEA..”

Compart SC-HEA and GB-HEA, when the strain rate increases from 5 × 108 s−1 to 5 × 109 s−1, the strain (ε) of GB-HEA is mostly higher than that of SC-HEA.

Reviewer 2 Report

The article was carefully reviewed and it may be considered for acceptance to publish after making the following corrections

The novelty of the work is good. However, quality of the figure 1, 2 and 5 must be improved, it is not clear to read it

Remaining things are in acceptable

Author Response

Response to Reviewer 2 Comment

The article was carefully reviewed and it may be considered for acceptance to publish after making the following corrections

The novelty of the work is good. However, quality of the figure 1, 2 and 5 must be improved, it is not clear to read it

Remaining things are in acceptable 

Response :

  All figures including figure 1, 2 and 5 have been revised carefully according to the requirement of this journal.

Reviewer 3 Report

Dear Authors,

The topic that you are investigating is very interesting and relevant. However, the manuscript needs extensive revision.

1)    English language and style must be edited, for instance, please change the word “stain” by strain in line 97.

2)    The introduction must be rewritten. The discussion introduced is poor, and more references must be added. The discussion on the factors that affect the crack propagation is really marginal and irrelevant, and it does not provide any reference about this issue.

3)    The chapter “Computational Methods” needs to be rewritten in a proper way, for instance, the way that equations are presented from line 64 to 68 should be rewritten in a way easier to interpreter, and  all the variables have to be defined.

4)    In the same chapter 2, the author mentions the standard ISO without referring a specific standard. Please specify and reference the specific ISO standard. On the other hand, ISO standard is not valid at the nanoscale. Therefore, it does not make sense to try to implement this standard. In addition, it is not possible to implement a perfectly built notch with an angle of 45°, because at the nanoscale the notch is built removing entire atoms, and it is not possible to remove just a small piece of any atom to obtain an perfect notch angle of 45°.

5)    It is not clear how the GB-HEA model is built. Please specify how the grain boundary is simulated in the models (What are the boundary conditions and all the details about how was the grain boundary represented?). In the figures 1 d and e, the grain boundary is not observed. Please, provide figures where the GB can be observed and the interaction between GB and dislocation.

6)    Provide the tilt angle of the misorientation grain boundry.

7)    It would be a good improvement to contrast the results in Table 1 with already published data from other authors.

Author Response

Response to Reviewer 3 Comments

The topic that you are investigating is very interesting and relevant. However, the manuscript needs extensive revision.

Point 1: English language and style must be edited, for instance, please change the word “stain” by strain in line 97.

Response 1:

We have edited carefully English language and style for this manuscript. We try our best to improve our manuscript’s quality.

Point 2:The introduction must be rewritten. The discussion introduced is poor, and more references must be added. The discussion on the factors that affect the crack propagation is really marginal and irrelevant, and it does not provide any reference about this issue.

Response 2:

We revised English language and added the references for the manuscript. We rewritted the introduction mainly as following.

Line 30-32 “The excellent mechanical properties of AlxCrCuFeNi HEA have attracted a large number of scholars to study deeply because of its remarkable mechanical strength, oxidation resistance and fatigue resistance at high temperature [2,3]..”

Line 37-38” The difficulty of preparation for AlxCrCuFeNi HEA prevented people from moving forward and the mechanism of plastic deformation are still unclear.”

Line 56-60” Li investigated Co25Ni25Fe25Al7.5Cu17.5 HEAs by a series of molecular-dynamics tensile tests at different strain rates and temperatures [10]. Wang studied the mechanical behaviors and deformation mechanisms of scratched AlCrCuFeNi HEAs by molecular dynamics simulations [11].”

Point 3:The chapter “Computational Methods” needs to be rewritten in a proper way, for instance, the way that equations are presented from line 64 to 68 should be rewritten in a way easier to interpreter, and  all the variables have to be defined.

Response 2:

We revised the line 75-79 in our manuscript. “

                                (1)

Where, Ei is the sum of embedded energy and counter potential of two systems; α and β represent the element types of atom i and j; rij is the distance between atom i and j; F represents the embedding energy function, which is the function of the electron density of all atoms in the system except itself ρ_; Φαβ represents the pair potential interaction between elements α and β, which is a function of the distance rij between atoms i and j.”

Point 4:In the same chapter 2, the author mentions the standard ISO without referring a specific standard. Please specify and reference the specific ISO standard. On the other hand, ISO standard is not valid at the nanoscale. Therefore, it does not make sense to try to implement this standard. In addition, it is not possible to implement a perfectly built notch with an angle of 45°, because at the nanoscale the notch is built removing entire atoms, and it is not possible to remove just a small piece of any atom to obtain an perfect notch angle of 45°.

Response 4:

We revised the line 85-87 in our manuscript.

“The standard in international standard organization is not valid at the nanoscale and therefore the angle pre-crack may be random [17]. In this paper, 45° is chosen. ”

At the same time, Figure 1 is revised.

Point 5: It is not clear how the GB-HEA model is built. Please specify how the grain boundary is simulated in the models (What are the boundary conditions and all the details about how was the grain boundary represented?). In the figures 1 d and e, the grain boundary is not observed. Please, provide figures where the GB can be observed and the interaction between GB and dislocation.

Response 5:

GB-HEA model is built as following. Compared with the SC-HEA, GB-HEA including grain boundary (GB) contains cluster microstructure, which is established based on the SC-HEA. SC-HEA was uniformly heated from 300 K to 2400 K, and then relaxed at 2400 K in order to make all atoms to keep liquid state. After that the Al0.1CoCrFeNi HEA was cooling down to room temperature at a certain cooling rate to form the cluster microstructure shown in Figure 1(d,e). In this situation, white atom forms subgrain boundary or grain boundary.

Point 6:Provide the tilt angle of the misorientation grain boundry.

Response 6:

In this paper, subgrain boundary or grain boundary in GB-HEA would be formed during solidification, therefore, the tilt angle is difficult to determine.

Point 7:It would be a good improvement to contrast the results in Table 1 with already published data from other authors.

Response 7:

There are few articles on the crack propagation of Al0.1CoCrFeNi HEA. Zhang[8] researched the local cracks on the surface and inside of HEA with FCC crystal structure, which gave us the research ideas. We also consulted and learned studies on crack expansion of other alloys and got a lot of ideas. Because Young’s modulus and yield strength of SC-HEA at different strain rates (Table 1) are simulated under the situation of the pre-crack, the comparison between SC-HEA and GB-HEA is meaningful. It is less meaningful to compare the values of Young’s modulus and yield strength with that of other HEAs.

Round 2

Reviewer 3 Report

I am satisfied with the modifications. Now, the manuscript is ready for being accepted.